# Radiation Dose Assessment for Myocardial Perfusion Imaging: A Single Institution Survey

Essam Alkhybari [1,*], Salman Albeshan [2], Bandar Alanazi [3], Raghad Alfarraj [4], Rakan Alduhaim [4], Intidhar El Bez Chanem [4] and Rima Tulbah [4]

1 Department of Radiology and Medical Imaging, College of Applied Medical Sciences, Prince Sattam Bin Abdulaziz University, Alkharj 11942, Saudi Arabia
2 Department of Radiological Sciences, College of Applied Medical Sciences, King Saud University, Riyadh 11433, Saudi Arabia
3 Nuclear Medicine Department, King Khalid Hospital, Hail Health Cluster, Hail 55421, Saudi Arabia
4 Nuclear Medicine Department, King Fahad Medical City, Riyadh 11525, Saudi Arabia
* Correspondence: e.alkhybari@psau.edu.sa

**Abstract:** Objective: This study aims to establish a local diagnostic reference level (LDRL) for single-photon emission tomography/computed tomography (SPECT/CT) and positron emission tomography/CT (PET/CT) with respect to myocardial perfusion imaging (MPI). Materials and Methods: The acquisition protocol and dosimetry data on the MPI procedures of five SPECT/CT scans and one PET/CT scan were collected. Data on technitum-99m sestamibi ($^{99m}$Tc-sestamibi), $^{99m}$Tc-tetrofosmin, thallium-201 ($^{201}$Tl), and rubidium-82 ($^{82}$RB) were all collected from one centre via questionnaire booklets. Descriptive data analysis was used to analyse all variables, and the 50th percentile was used to analyse each radiation dose quantity. Results: The reported 50th percentile dose for a one-day stress/rest protocol using $^{99m}$Tc-sestamibi (445/1147 MBq) and $^{99m}$Tc-tetrofosmin (445/1147 MBq) and for a two-day stress/rest protocol using $^{99m}$Tc-sestamibi (1165/1184 MBq) and $^{99m}$Tc-tetrofosmin (1221/1184 MBq) are in good agreement with reported national diagnostic reference levels (NDRLs). However, the dose from the study data on a one-day stress/rest protocol using $^{99m}$Tc-sestamibi was more than the 50th percentile dose from the Brazilian data (370/1110 MBq) on a similar protocol, and the dose from the study data on a two-day stress/rest protocol using $^{99m}$Tc-tetrofosmin was more than the 50th percentile dose (1084/1110 MBq) from the United States data on MPI scans. Regarding the computed tomography (CT) portion of the SPECT/CT framework, the 50th percentile doses were lower than all the identified doses in the data considered in the literature reviewed. However, regarding the CT component of the PET/CT MPI scans, the $^{82}$RB dose was more than the recorded doses in the CT data in the published literature. Conclusion: This study determined the LDRL of five SPECT/CT protocols and one PET/CT MPI protocol. The results suggest that there may be opportunities to optimise the patient radiation burden from administered activities in patients undergoing SPECT examinations and the CT components associated with $^{82}$RB PET/CT scans without compromising diagnostic image quality.

**Keywords:** SPECT/CT; PET/CT; LDRL; diagnostic reference levels

## 1. Introduction

Globally, cardiovascular diseases are the leading cause of morbidity and mortality, with a 17.9 million annual death rate, including 9.4 million from coronary artery disease (CAD) [1]. Imaging techniques such as myocardial perfusion imaging (MPI), single-photon emission computed tomography/computed tomography (SPECT/CT), and positron emission tomography/CT (PET/CT) play an indispensable role in the diagnosis and treatment of patients with a cardiovascular disease (CVD) [2]. Over the past two decades, the number of MPI procedures performed worldwide reached 20 million annually. The proliferation of

SPECT/CT and PET/CT technology and expertise has engendered the growing adoption of these technologies across developing countries [2,3].

The MPI protocols involve exposing the patient to an internal radiation dose of nuclear medicine (NM) or PET radiopharmaceuticals and an external radiation dose for computed tomography (CT), which is used for attenuation correction (AC) or anatomical localisation (AL), or both AC and AL (AC-AL), or diagnostic CT [4,5]. Notwithstanding the benefits of stress MPI, there is growing public concern over the risks of ionising radiation from PET/CT and SPECT/CT MPI procedures. In the United States, current published data presented by Einstein et al. (2015) [6] and Jerome et al. (2015) [3] pointed out that a large number of United States SPECT/CT and PET/CT departments continue to administer radiopharmaceuticals at higher doses than the recommended MPI radiation doses. Findings from these data reveal that adherence to NM and molecular imaging best practices for MPI may be suboptimal across US facilities [3,6] In response to the growing public concern, the radiation dose for MPI scans should be optimised, such that the patient receives the smallest radiation dose that would yield the appropriate diagnostic image quality [7].

In clinical practice, a wide range of SPECT/CT and PET/CT protocols and radiopharmaceuticals can be used to perform the MPI scan while ensuring that a lower radiation dose is used on the patient [1,2]. The cardiac SPECT/CT and PET/CT MPI scan commonly consist of the stress protocol, which involves treadmill exercise or vasodilator drugs, and the rest protocol. The most common SPECT/CT MPI protocol conducted in one day, either starting with stress or rest, or two-day protocols uses $^{99m}$Tc-Tetrofosmin, $^{99m}$Tc-Sestamibi, or thallium-201 ($^{201}$Tl). In some circumstances, the patient undergoes only a stress SPECT/CT MPI and the stress result becomes normal; therefore, there is no need to do the rest protocol. Notably, doing the stress and rest MPI SPET/CT scan resulted in more radiation burden than doing only the stress SPECT MPI protocol [1,2,5]. Moreover, $^{201}$Tl is a good tracer for MPI; however, it has numerous limitations, such as a long half-life, high radiation exposure to the patient, relatively low injected activity, low signal-to-noise ratio, and, in some cases, suboptimal images particularly in overweight patients [1,2,5].

Indeed, the primary challenge for MPI procedures is designing an MPI protocol that delivers the lowest radiation burden necessary to acquire adequate diagnostic image quality via NM administered activity during SPECT or PET and the CT component of SPECT/CT and PET/CT scans [7,8]. To facilitate the dose reduction process, the concept of diagnostic reference levels (DRLs) has emerged as a gauge for monitoring the radiation doses of diagnostic medical imaging modalities. The DRLs in NM and molecular imaging describe dose levels for administered activities for procedure-specific radiopharmaceuticals and are expressed in millicuries (mCi) or megabecquerels (MBq) and volume CT dose index (CTDI$_{vol}$), and in dose length product (DLP) (mGy.cm) for the CT component associated with NM and molecular imaging procedures during SPECT and PET examinations [7,9]. The recommended value for each radiation dose quantity is derived based on the third quartile (75th percentile) of the national DRL (NDRL) audit or the 50th percentile of the corresponding local DRL audit. To determine the radiation dose, the NM or molecular PET imaging centres should compare their median radiation dose against the NDRL of their country. In the absence of an NDRL, the derived median radiation dose, which represents the 50th percentile, was compared with other values from published NDRL and LDRL studies [7,9].

MPI scans are classified as one of the top contributors to the radiation burden of patients, which indicates an urgent need to establish DRLs for SPECT/CT and PET/CT dose reduction strategies across the globe [10]. Indeed, derivation and utilisation of the NDRL method in MPI SPECT/CT and PET/CT lag behind other fields such as diagnostic CT. A DRL audit implemented by the European Commission reports that a comparison of DRLs for MPI procedures is challenging because of discrepancies in the reporting of MPI protocols between the participating developed European countries and the rest of the world [10]. This is because the majority of NDRL reports focus on reporting administered activity from only MPI SPECT or MPI PET scans, with only a few identified studies reporting

NDRLs for the CT component of these examinations [9]. Furthermore, the literature review highlighted that variation in NDRL value for the administered activities related to the stress and rest one-day protocol range from 407 MBq and 740 MBq to 555 MBq and 1184 MBq for $^{99m}$Tc-Tetrofosmin /Sestamibi, respectively [8,11,12]. Regarding CT components, a few articles reflect the change of variation in NDRL related to the CT dose, for instance, the CTDI$_{vol}$ value ranges from 2.2 mGy to 36 mGy and the DLP value ranges from 36 mGy.cm to 380 mGy.cm [13–17].

NM services in Saudi Arabia are provided by many health sector establishments. In 2017, there were 21 PET/CT scanners, 55 SPECT/CT scanners, and 35 gamma camera SPECT systems at healthcare facilities across the country [18]. Approximately 12,387 MPI NM scans were implemented in 2017 at multiple NM departments and centres spread over 13 administrative areas [18]. However, published NDRL or LDRL methods for MPI SPECT/CT and PET/CT scans in Saudi Arabia are not yet available. The objective of this study is to determine the LDRL for MPI SPECT/CT and PET/CT scans and to compare the derived dose against the 50th percentile and 75th percentile doses in published NDRL methods.

## 2. Materials and Methods

The study recruited all adult Saudi patients who suffered from CVD and were eligible to undergo SPECT/CT or PET/CT MPI scans between 2020 and 2021. Ethical approval (IRB log number: 21-204E) was acquired before commencing the study. The data were gathered from scans conducted on one SPECT/CT scanner and one PET/CT scanner in Riyadh, Saudi Arabia. Five SPECT/CT MPI scans were selected for this study: one-day stress/rest ($^{99m}$Tc-Sestamibi and $^{99m}$Tc-Tetrofosmin), two-day stress/rest ($^{99m}$Tc-Sestamibi and $^{99m}$Tc-Tetrofosmin), and stress/rest and reinjection $^{201}$Tl. For PET/CT MPI scans, one stress/rest rubidium-82 ($^{82}$RB) was selected for this study. A retrospective data collection approach was implemented to collect all the required information for this project. Two NM technologists were assigned to collect all the data-based booklet questionnaires used in this study. All the required data had been registered on hospital radiology systems and picture archiving communication systems (PACS).

The first part of the data collection aspect of this study focused primarily on obtaining information on the MPI protocol used at the study facility, such as the model and brand name of the SPECT/CT and PET/CT scanners, the manufacturer, commission date, detector material, and recommended administration method for radioactivity. The second part of the LDRL audit focused on collecting individual demographic data and radiation dose information on each patient who underwent the MPI SPECT/CT and PET/CT protocols considered in this study. The demographic data include the clinical indication, gender, age, weight, and height. Radiation dose information includes the type of administered activity during each SPECT and PET examination and CT dose quantity in CTDI$_{vol}$ and DLP. Details on the acquisition time, matrix size, and collimator were collected for SPECT MPI scans, while scanning time and number of bed/positions was collected for PET MPI scans. CT parameters in the axillary of SPECT and PET MPI were collected, including tube rotation time, kilovoltage, scan length, pitch, and slice thickness. The data collection approach followed the non-weight restricted protocol consistent with the NDRL method for adult common SPECT/CT and PET/CT scans used in a published international study implemented by the Australian Radiation Protection and Nuclear Safety Agency [19].

All the CT data related to the SPECT/CT MIP scans were used for AC, and the CT portion of the PET/CT scan was used for AC-AL. The CT equipment was calibrated by the manufacturer using a 32 cm body phantom for CT components associated with all identified SPECT MPI protocols and the PET MPI protocol. Automatic exposure control software was utilised for all identified CT MPI acquisition protocols. The scanner installation data were collected in June 2012 for the SPECT/CT scanner and 2017 for the PET/CT scanner.

Regarding the statistical data analysis, all the collected data were transferred to an Excel sheet. Then, after data entry, they were transferred to SPSS software version 18.0 (PASW, Chicago, IL, USA). The LDRL value for administered activity, CTDI$_{vol}$, and

DLP in frameworks of SPECT/CT and PET/CT MPI were determined based on 50th percentiles. Moreover, the SPECT, PET, and CT data were analysed using descriptive statistics, such as mean, minimum, maximum, and standard deviation (SD). The SPECT and PET radiopharmaceuticals data were analysed and compared with the 50th and 75th percentiles of international published MPI data. Likewise, the CT components associated with SPECT/CT and PET/CT MPI scans were analysed and compared with 50th and 75th percentiles of published international NDRL studies.

## 3. Results

Table 1 summarises the descriptive statistical analysis results, demonstrating the number of patients, percentage of males and females, and SD for the age and weight of all identified SPECT/CT and PET/CT MPI protocols. It presents 500 MPI cases from four $^{99m}$Tc SPECT/CT MPI scans, 57 cases from one $^{201}$TL SPECT/CT scan, and 49 cases from one $^{82}$RB PET/CT MPI scan. The statistical data analysis of the LDRLs for the SPECT/CT MPI protocols and the PET/CT MPI protocol are described in Table 2. The aggregated LDRLs for this study, employing $^{99m}$Tc-Sestamibi for a one-day stress (low)/rest (high) protocol for administered activity, were 12.30 mCi (455.10 MBq) and 31 mCi (1147 MBq). The LDRLs for the stress (low)/rest (high) administered activity for the $^{99m}$Tc-Tetrofosmin protocol were 12.23 mCi (455.10 MBq) and 31 mCi (1147 MBq). For the two-day protocols, the LDRLs for the administered activity on the first and second days were 31.50 (1165.50) and 32 mCi (1184 MBq), and 33 mCi (1221 MBq) and 32 mCi (1184 MBq), respectively, for $^{99m}$Tc-Sestamibi and $^{99m}$Tc-Tetrofosmin. The derived LDRLs for the stress/rest and re-injection administered activity for the $^{201}$Tl-201 MPI protocol were 3.5 mCi (129.5 MBq) and 1 mCi (37 MBq). For the $^{82}$RB MPI PET/CT protocol, the determined LDRL for the stress/rest protocol was 25 mCi (925 MBq). The derived LDRLs for the CT portion in a framework of diverse MPI protocols are presented in Table 3.

One SPECT/CT system (Discovery NM/CT 670) and one PET/CT system (GE Discovery IQ 5-Ring) were identified during the LDRL audit. The NM department details on the MPI acquisition protocol for an MPI one-day or two-day protocol ($^{99m}$Tc-tetrofosmin or $^{99m}$Tc-sestamibi) specify using a large-field gamma camera with a low-energy high-resolution (LEHR) collimator, a 64 × 64 matrix size, a zoom of 1.3, a $^{99m}$Tc peak at 140 keV, and a 20% window. The acquisition time for the stress protocol was 13 min, and for the rest protocol, it was 10 min, whereas the acquisition time for the two-day protocol was 10 min for stress or rest. For the $^{201}$Tl MPI protocol, the equipment used included a large-field gamma camera with an LEHR collimator, a 64 × 64 matrix size, a zoom of 1.3, a $^{201}$Tl peak at 70 keV and a 15% window, and at 167 keV and a 10% window. The acquisition time for stress, rest, and delayed images (24 h) was 15 min. The CT components within the framework of SPECT acquisition parameters ($^{99m}$Tc-tetrofosmin, $^{99m}$Tc-sestamibi, and $^{201}$Tl) were 120 kVp, tube current−time of 20 mAs, and a pitch ratio of 0.93. For $^{28}$RB MPI, the PET protocol used was one bed in 6 min. The CT components within the framework of PET acquisition parameters were 120 kVp, 60 mAs, slice thickness of 3.75 mm, and pitch ratio 1.37.

**Table 1.** Patient demographics data for the common myocardial perfusion imaging (MPI) protocols.

| Characteristics | SPECT MPI Protocol | | | | | PET MPI Protocol |
|---|---|---|---|---|---|---|
| | 1-Day Stress/Rest $^{99m}$Tc-Sestamibi | 1-Day Stress/Rest $^{99m}$Tc-Tetrofosmin | 2-Day Stress/Rest $^{99m}$Tc-Sestamibi | 2-Day Stress/Rest $^{99m}$Tc-Setrofosmin | Stress/Rest $^{201}$TL | Stress/Rest $^{82}$RB |
| Number of patients | 100 | 100 | 100 | 99 | 57 | 49 |
| Number of males (%) | 52 (52%) | 57 (57.6%) | 41 (41%) | 48 (48.5%) | 33 (56.9%) | 17 (34.7%) |
| Number of females (%) | 48 (48%) | 42 (42.4%) | 59 (59%) | 51 (51.5%) | 24 (41.4%) | 32 (65.3%) |
| Patient age, years (mean, SD) | 60.46 (±11.60) | 62.85 (±13.23) | 58.89 (±14.17) | 59.36 (±14.12) | 58.24 (±14.11) | 57.3 (±13.46) |
| Patient weight, kg (mean, SD) | 75.22 (±13.63) | 75.7 (±12.27) | 88.05 (±19.25) | 89.51 (±18.85) | 86.75 (±19.41) | 107.56 (±18.81) |

**Table 2.** Radiation dose measurements for the common administered activities delivered from diverse myocardial perfusion imaging protocols.

| MPI Protocol | Radiopharmaceutical | Administered Activity mCi (MBq) | | | |
|---|---|---|---|---|---|
| | | Mean | Min | Max | DRL (50th Percentile) |
| One-day stress/rest | 1st (Low A.A) $^{99m}$Tc-sestamibi | 12.24 (452.95) | 11 (407) | 14 (518) | 12.30 (455.10) |
| | 2nd (High A.A) | 31.2 (1156.2) | 29 (1073) | 33 (1221) | 31 (1147) |
| One-day stress/rest | 1st (Low A.A) $^{99m}$Tc-tetrofosmin | 12.23 (452.85) | 11 (407) | 14 (518) | 12.23 (455.10) |
| | 2nd (High A.A) | 31.24 (1155.97) | 29 (1073) | 33 (1221) | 31 (1147) |
| Two-day stress/rest | 1st A.A $^{99m}$Tc-sestamibi | 31.94 (1181.78) | 30 (1110) | 37 (1369) | 31.50 (1165.50) |
| | 2nd A.A | 32.06 (1186.22) | 30 (1110) | 38 (1406) | 32 (1184) |
| Two-day stress/rest | 1st A.A $^{99m}$Tc-tetrofosmin | 31.92 (1181.01) | 30 (1110) | 37 (1369) | 33 (1221) |
| | 2nd A.A | 32 (1184) | 30 (1110) | 38 (1406) | 32 (1184) |
| Stress/rest | $^{201}$TL | 3.5 (129.5) | 3.5 (129.5) | 3.5 (129.5) | 3.5 (129.5) |
| Reinjection $^{201}$TL | 201TL redistribution | 1 (37) | 1 (37) | 1 (37) | 1 (37) |
| One-day stress/rest | 1st A.A $^{82}$RB | 24.90 (921.37) | 20 (740) | 25.10 (928.70) | 25 (925) |
| | 2nd A.A | 24.91 (921.67) | 20 (740) | 25.5 (943.50) | 25 *(925) |

Abbreviation: A.A; administered activity.

**Table 3.** Radiation dose measurements for the CT portion in framework of diverse myocardial perfusion imaging protocols.

| MPI Protocol | CT Radiation Dose Quantities | |
|---|---|---|
| | CTDI$_{vol}$ (mGy) | DLP (mGy.cm) |
| | LDRL (50th) | |
| One-day stress/rest, $^{99m}$Tc-sestamibi | 1.03 | 24.30 |
| One-day stress/rest, $^{99m}$Tc-tetrofosmin | 1.03 | 24.23 |
| Two-day stress/rest, $^{99m}$Tc-sestamibi | 1.03 | 24.23 |
| Two-day stress/rest, $^{99m}$Tc-tetrofosmin | 1.03 | 24.23 |
| Stress/rest 201TL | 1.03 | 23.84 |
| One-day stress/rest, $^{82}$RB | 7.32 | 207.76 |

Table 4 presents a comparison of the LDRLs for SPECT and PET MPI imaging scans from our study against those in the recognised published studies on MPI NDRLs. Table 5 presents a comparison of the data on LDRLs for MPI from our study against the data from recognised published studies on MPI NDRLs for the CT component of SPECT/CT and PET/CT MPI scans.

**Table 4.** Comparison between the LDRL for MPI imaging from our study against the recognised MPI international studies for NM and PET administered activities.

| NO | Protocol | | LDRL Project | Becker et al. (USA, 2016) [11] | | Song et al. (KO, 2019) [12] | | Willegaignon et al. (BR, 2015) [20] | | Hirschfeld et al. (INCAPS, 2021) [8] | |
|---|---|---|---|---|---|---|---|---|---|---|---|
| | | | 50th | 50th | 75th | 50th | 75th | 50th | 75th | 50th | 75th |
| 1 | **1-day stress/rest** $^{99m}$**Tc-MIBI** | 1st dose | 455.10 (12.30) | 388 (10.50) | 425 (11.50) | 555 - | 740 - | 370 - | 444 - | 374 (10.10) | 414 (11.20) |
| | | 2nd dose | 1147 (31.00) | 1169 (31.60) | 1251 33.80 | 925 - | 1110 - | 1110 - | 1110 - | 1036 (28.00) | 1184 (32.20) |
| 2 | **1-day stress/rest** $^{99m}$**Tc-Tetro \*** | 1st dose | 452.85 (12.23) | 388 (10.50) | 425 11.5 | 555 - | 740 - | 370 - | 444 - | 374 (10.10) | 414 (11.20) |
| | | 2nd dose | 1155.97 (31.24) | 1147 (31.00) | 1221 33.00 | 925 - | 1110 - | 1110 - | 1110 - | 1036 (28.00) | 1184 32.00 |
| 3 | **2-day stress/rest** $^{99m}$**Tc-MIBI** | 1st dose | 1165.50 (31.50) | 1089 (29.40) | 1165 31.50 | - | - | 740 | 870 | 657 (17.80) | 851 (23.00) |
| | | 2nd dose | 1184 (32.00) | 1110 (30.00) | 1184 32.00 | - | - | 814 | 925 | 690 (18.70) | 666 (18.00) |
| 4 | **2-day stress/rest** $^{99m}$**Tc-Tetro** | 1st dose | 1221 (33.00) | 1084 (29.30) | 1214 32.80 | - | - | 740 | 870 | 657 (17.80) | 851 (23.00) |
| | | 2nd dose | 1184 (32.00) | 1110 (30.00) | 1199 32.40 | - | - | 814 | 925 | 690 18.70 | 666 18.00 |
| 5 | **Stress/rest** $^{201}$**TL Redistribution** | 1st dose | 129.5 (3.50) | - | - | - | - | 111 | 130 | 111 3.00 | 111 3.00 |
| | | 2nd dose | 37 (1.00) | - | - | - | - | - | - | 38 1.00 | 41 1.10 |
| 6 | **1-day stress/rest** $^{82}$**RB** | 1st dose | 925 (25.00) | - | - | - | - | - | - | - | - |
| | | 2nd dose | 925 (25.00) | - | - | - | - | - | - | - | - |

Abbreviation: USA; United States of America, KO; Korea, BR; Brazil, and INCAPS; IAEA Nuclear Cardiology Protocols Study.

**Table 5.** Comparison between the LDRL for MPI imaging from our study against the recognised MPI international studies for CTDI$_{vol}$ and DLP in framework of MPI SPECT/CT or PET/CT scan.

| Authors | Modality | Radiotracer | Clinical Purpose | MPI CT DRL Values | | | |
|---|---|---|---|---|---|---|---|
| | | | | CTDI$_{vol}$ (mGy) | | DLP (mGy.cm) | |
| **DRL (Statistical Analysis)** | | | | 50th | 75th | 50th | 75th |
| Rinscheid et al. (DE, 2022) [13] | SPECT/CT and PET/CT | $^{99m}$Tc and $^{18}$F-FDG | AC | - | 3.1 | - | 81 |
| Bebbington et al. (Nordic, 2019) [14] | SPECT/CT and PET/CT | $^{99m}$Tc and $^{18}$F-FDG | | 1.6 | 2.2 | 35 | 53 |
| Iball et al. (UK, 2017) [15] | SPECT/CT | $^{99m}$Tc | AC | 1.6 | 2.1 | 34 | 36 |
| Abe et al. (JP, 2020) [16] | SPECT/CT | $^{9m}$Tc | AC+Dx | 3.2 | 4.50 | 89 | 180 |
| Abe et al. (JP, 2020) [16] | PET/CT | $^{18}$F-FDG | Cl+Dx | 5.5 | 9.10 | 209 | 380 |
| Lima et al. (Swiss, 2018) [17] | SPECT/CT and PET/CT | $^{99m}$Tc and $^{18}$F-FDG | AC-AL | 2 | 40 | 1 | 10 |
| This project LDRL | SPECT/CT | $^{99m}$Tc | AC | 1.03 | | 24.33 | |
| This project LDRL | PET/CT | $^{82}$RB | AC-AL | 7.32 | | 207.76 | |

Abbreviation: DE; Germany, UK; United Kingdom; JP; Japan; Swiss; Switzerland, Cl; clinical, and Dx; diagnostic CT.

## 4. Discussion

This is the first adult SPECT/CT and PET/CT dose survey, with LDRLs derived for several MPI SPECT/CT scans and one PET/CT scan, in Saudi Arabia (Table 2). There are many classes of DRLs depending on the target population of the dose survey, including LDRL, regional DRL (RDRL), and NDRL. The scope of this study is consistent with the LDRL method, and our recommended values were compared against published 50th percentile doses and NDRLs (75th percentile doses). This approach aids in the identification

of where there is a need to optimise radiation doses, which is when the median derived value is greater than the NDRL in the published data [7]. In essence, optimising radiation doses becomes necessary when the median dose exceeds the NDRL. Comparing our 50th percentile dose from the SPECT/CT MPI protocols and the PET/CT MPI protocol with the 50th percentile doses of other published NDRL methods will facilitate further optimisation without deterioration of the diagnostic image quality of SPECT/PET images and their CT components. However, it should be noted that there will be a variance between the recommended radiation doses in our study and the doses in other published data. Thus, a data comparison should consider differences in patient body habitus, administered activity/weight (MBq/kg), equipment, NM physician experience, and department workflow.

Table 2 shows the radiation dose measurements for the typical NM and PET administered activities delivered from diverse MPI imaging protocols. The reported 50th percentile for the stress and rest SPECT MPI protocols associated with both stress and rest protocols using $^{99m}$Tc-sestamibi and $^{99m}$Tc-tetrofosmin radiopharmaceuticals were more than the 50th and 75th percentile doses of the recognised published stress and rest MPI NDRL methods (Table 4). The variance between the 50th percentile doses for a one-day stress MPI $^{99m}$Tc-sestamibi scan (445.10 MBq) in our study and the 50th percentile dose (374 MBq) and 75th percentile dose (414 MBq) in the data from the study by Hirschfeld et al. (2021) [1] is approximately $\times 1.21$ and $\times 1$, respectively. For a two-day MPI SPECT scan, the determined 50th percentile doses for a stress and rest protocol administering $^{99m}$Tc-sestamibi and $^{99m}$Tc-Tetrofosmin in this study were more than the 50th and 75th percentile doses reported as NDRLs for Brazil in 2015, those reported in the international survey of the 2021 IAEA Nuclear Cardiology Protocols Study (INCAPS) project, and the 75th percentile doses reported for France in 2017 (Table 4). Understanding the disparities in MPI stress and rest administered activities is complicated by differences in the data collection methods, e.g., administered activity/weight vs. fixed administered activity, different weight restrictions (70 $\pm$ 10 kg) vs. no weight restriction approach [1,9]. However, optimising the administered activity for $^{99m}$Tc-Sestamibi and $^{99m}$Tc-Tetrofosmin one-day or two-day protocols is indispensable for achieving appropriate diagnostic image quality.

Regarding the data on the $^{201}$Tl stress/rest and redistribution procedure and the one-day stress/rest $^{82}$RB PET/CT MPI procedure, a few studies have reported NDRLs for the protocol for $^{201}$Tl stress/rest MPI SPECT/CT scan and redistribution (second injection) after 24 h [1,21]. The determined LDRLs for a stress/rest $^{201}$Tl SPECT/CT scan in this study are more than the 50th and 75th percentile doses for France and the INCAPS NDRL projects (Table 4). Although the details of the NDRL data collection methods are inconsistent, the reason for the variance in $^{201}$Tl administered activities might stem from the administered activity/weight, the MPI acquisition protocol, or the scanner technology. However, the LDRLs for redistribution injection of $^{201}$Tl for a SPECT/CT MPI scan were lower than the NDRLs for France and Brazil, and the INCAPS values [1,20,21]. Regarding the $^{82}$RB PET/CT MPI scan, there is no available NDRL method or LDRL method with reported DRLs for a one-day stress/rest $^{82}$RB PET/CT scan. The derived LDRL values in this study for each administered activity associated with a one-day stress/rest $^{82}$RB SPECT/CT MPI scan were lower than the recommended values for stress/rest administered activity stipulated by the international guideline (2220 MBq) [22].

In the literature reviewed, we highlighted that a few published NDRL studies have reported NDRLs for the CT portion of $^{99m}$Tc SPECT/CT and fluoride-18 fluorodeoxyglucose ($^{18}$F-FDG) PET/CT scans, with the CT purposely used for AC, AC-AL, or AC with diagnostic CT. The participating NM and PET/CT departments used the same CT acquisition parameter for all identified CT protocols associated with $^{99m}$Tc-Sestamibi, $^{99m}$Tc-tetrofosmin, and $^{201}$Tl SPECT/CT MPI scans. Consequently, there is no variation in the reported LDRLs for the CTDI$_{vol}$ and DLP for each of the considered SPECT/CT MIP scans (Table 3). In this study, the 50th percentile CTDI$_{vol}$ and DLP values were lower than the reported 50th and 75th percentile NDRL values in recognised published international data (Table 5). Although

the CT parameters used in this study were 20 mA and 120 kVp, no published NDRL data include details on CT acquisition protocols, which would facilitate an understanding of the reasons behind differences in CT radiation doses.

The CT radiation doses delivered for a one-day stress/rest $^{82}$RB PET/CT MPI scan were compared with the available published NDRL data on combined CT from $^{99m}$Tc SPECT/CT $^{18}$F-FDG PET/CT MPI scans (Table 3). It is noteworthy that the derived 50th percentile dose for the CT portion within the framework of the $^{82}$RB PET/CT procedure was more than other published CT doses used for AC, AC-AL, and diagnostic CT, including those for Germany in 2022, the Nordic countries in 2019, Japan in 2020, and Switzerland in 2018 [18–21]. However, it is difficult to precisely decipher the probable reason for disparities in CTDI$_{vol}$ and DLP values across different NDRL methods vis-à-vis those in our study because of the unavailability of details on CT acquisition parameters. However, the tube current−time value reported for in PET/CT protocols was 60 mAs, compared to the 20 mAs for a $^{99m}$Tc SPECT/CT MPI scan in all the considered SPECT/CT protocols in our study. However, there is a need to implement a dose-reduction strategy to reduce the radiation doses of the CT portion within the framework of an $^{82}$RB PET/CT scan.

## 5. Limitations

This study has some limitations. Although the reported MPI radiation dose quantities are based on standard body weight patients, and the majority of patients involved in this study weighed more than standard-sized patients, it is necessary to take into account the reported DRLs for radiation doses delivered for MPI scans when the standard body weight is different, as it is highly likely that the standard weight for the West differs from that for the Middle East. Another limitation is that the derived LDRLs for MPI scans were obtained from only one participant PET/CT centre. Thus, the applicability of the LDRL values derived in this research is questionable, as it represents the SPECT/CT and PET/CT centre of only one specific hospital instead of all SPECT/ CT and PET/CT centres in Saudi Arabia.

## 6. Conclusions

This study determined the LDRLs for five SPECT/CT scans, including a $^{99m}$Tc-Sestamibi and a $^{99m}$Tc-tetrofosmin stress and rest MPI scan, a $^{201}$Tl one-day stress/rest MPI scan, and a PET/CT scan from an $^{82}$RB stress/rest MPI scan. The values derived for a one-day and a two-day stress/rest SPECT/CT MPI scan protocol were in good agreement with the published NDRL methods. However, it is recommended to optimise all studied stress/rest SPECT/CT scan protocols for $^{99m}$Tc and $^{201}$Tl MPI radiotracers without compromising image quality. The derived LDRL values for a one-day stress/rest $^{82}$RB PET/CT MPI scan were lower than the reported international guidelines for stress/rest $^{82}$RB cardiac scans. Regarding the CT portion within the framework of a SPECT/CT MPI scan, the reported DLP and CTDI$_{vol}$ values for CT used for AC were lower than the published NDRLs recognised in the literature reviewed. However, the numerical LDRL value for CT associated with an $^{82}$RB PET/CT MPI scan was more than the combined CT for a $^{99m}$Tc and $^{18}$F-FDG PET scan reported in the literature. Consequently, there is a need to optimise CT for $^{82}$RB PET/CT scans while still achieving good image quality. The data on reported LDRLs for MBq, CTDI$_{vol}$, and DLP radiation doses in this study may be useful to Saudi Arabian SPET/CT and PET/CT centres for a comparison of their median typical LDRL values against the LDRL values published in this study. However, further research should be conducted to derive NDRLs for stress/rest SPECT/CT and PET/CT MPI scans.

**Author Contributions:** Conceptualisation, E.A., S.A., I.E.B.C. and R.T.; methodology, E.A., B.A. and R.A. (Raghad Alfarraj); resources, R.A. (Raghad Alfarraj), R.A. (Rakan Alduhaim), B.A., I.E.B.C. and R.T.; data curation, E.A.; writing—original draft preparation, E.A. and S.A.; writing—review and editing, E.A. and S.A.; visualisation, E.A., S.A. and I.E.B.C.; supervision, E.A. and R.T.; project administration, E.A. All authors have read and agreed to the published version of the manuscript.

**Funding:** This research received no external funding.

**Institutional Review Board Statement:** Ethical approval (IRB log number: 21-204E) was acquired before commencing the study from the participating centre.

**Acknowledgments:** This publication was supported by the Deanship of Scientific Research at Prince Sattam Bin Abdulaziz University, Alkharj, Saudi Arabia.

**Conflicts of Interest:** The authors declare no conflict of interest.

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
