# Peer review of "Radiation Dose Assessment for Myocardial Perfusion Imaging: A Single Institution Survey"

_tomography, doi:10.3390/tomography9010021_

Round 1

Reviewer 1 Report

In my opinion the manuscript is well-written. I would add some information regarding MPI protocols in the introduction, explaining better in particular stress-rest protocols. 

Author Response

Response to Reviewer 1 Comments

Point 1: In my opinion the manuscript is well-written. I would add some information regarding MPI protocols in the introduction, explaining better in particular stress-rest protocols. 

Response 1: Thanks a lot for providing your valuable comments. I have updated the introduction based on your comments. I have provided some explanations related to the stress and rest protocols, as follows: “In clinical practice, a wide range of SPECT/CT and PET/CT protocols and radiopharmaceuticals can be used to perform the MPI scan while ensuring that a lower radiation dose is used on the patient [1,2]. The cardiac SPECT/CT and PET/CT MPI scan commonly consist of the stress protocol, which involves treadmill exercise or vasodilator drugs, and the rest protocol. The most common SPECT/CT MPI protocol conducted in one day, either starting with stress or rest, or two days protocols uses 99mTc-Tetrofosmin, 99mTc-Sestamibi, or thallium-201 (201Tl). In some circumstances, the patient undergoes only a stress SPECT/CT MPI and the stress result becomes normal; therefore, there is no need to do the rest protocol. Notably, doing the stress and rest MPI SPET/CT scan resulted in more radiation burden than doing only the stress SPECT MPI protocol [1,2,5]. Moreover, 201Tl is a good tracer for MPI; however, it has numerous limitations, such as a long half-life, high radiation exposure to the patient, relatively low injected activity, low signal-to-noise ratio, and, in some cases, suboptimal images particularly in overweight patients [1,2,5]..”

Reviewer 2 Report

The study has patients from many ethnicity , age group is also variable which will effect result

SPECT/ CT.PET/CT which are used are of different make so that will effect the readings

Author Response

Response to Reviewer 2 Comments

Point 1: Comments and Suggestions for Authors. The study has patients from many ethnicity , age group is also variable which will effect result SPECT/ CT.PET/CT which are used are of different make so that will effect the readings

Response 1: Thanks for providing these valuable comments. I have to say that the data was collected only from Saudi patients who suffered from cardiovascular diseases and were eligible to undergo MPI SPECT/CT or PET/CT scans. I wrote a clear statement in the updated methodology section, as follows: “The study recruited adult Saudi patients who suffered from CVD and were eligible to undergo SPECT/CT or PET/CT MPI scans between 2020 and 2021.”

Moreover, the variation in radiation dose mainly from the weight of the patient. It is difficult to find a standard-sized patient (70 ± 10 kg) undergoing a cardiac MPI scan. The patient selection in this study adopted a non-weight-restriction approach to report the LDRL value. In the limitation section, we highlighted this issue as follows” Although the reported MPI radiation dose quantities are based on standard body weight patients, and the majority of patients involved in this study weighed more than standard-sized patients, it is necessary to take into account the reported DRLs for radiation doses delivered for MPI scans when the standard body weight is different, as it is highly likely that the standard weight for the West differs from that for the Middle East.”

Point 2: Is the researh deisng is approporiate:

Response 2: I have updated the introduction and methodology sections to make sure the design of methodology section reflects the gap that I need to address in this paper. I have added some sentences to reflect the changes in radiation doses in the published international data at the end of the introduction section, as follows: “The literature review highlighted that variation in NDRL value for the administered activities related to the stressand rest one-day protocol which range from 407 MBq and 740 MBq to 555 MBq and 1184 MBq for 99mTc-teterofosmine/Sestamibi, respectively. Regarding CT components, a few articles reflect the change of variation in NDRL related to the CT dose; for instance, the CTDIvol value ranged from 2.2 mGy to 36 mGy and the DLP value ranged from 36 mGy.cm to 380 mGy.cm.”

Point 3: Are the methods adequately desribed?

I have updated the methodology section as follows: “The study recruited all adult Saudi patients who suffered from CVD and were eligible to undergo SPECT/CT or  PET/CT MPI scans between 2020 and 2021. Ethical approval (IRB log number: 21-204E) was acquired before commencing the study.The data were gathered from scans conducted on one SPECT/CT scanner and one PET/CT scanner in Riyadh, Saudi Arabia. Five SPECT/CT MPI scans were selected for this study: one-day stress/rest (99mTc-Sestamibi and 99mTc-Tetrofmise), two-day stress/rest (99mTc-Sestamibi and 99mTc-Tetrofmise), and stress/rest and reinjection 201TL. For PET/CT MPI scans, one stress/rest 82RB was selected for this study.”

Point 4: Are the results clearly presented?

Response 5: I made some minor modifications to clarify the value of the SPECT/CT MPI protocols.

Point 5: Are the conclusions supported by the results?

Response 5: Thanks for providing this valuable comment. I think the conclusions are supported by results, as Reviewers 1 and 3 have indicated. Therefore, I will keep the conclusions as they are.

Reviewer 3 Report

The author should provide more literature/references since it is a survey.  

The author should also present the drawbacks/defects and advantages of single-photon-emission-tomography/computed tomography. 

Author Response

Response to Reviewer 3 Comments

Point 1: Comments and Suggestions for Authors. The author should provide more literature/references since it is a survey.  The author should also present the drawbacks/defects and advantages of single-photon-emission-tomography/computed tomography

Response 1: Thanks a lot for providing your valuable comments. I have searched through all the available published cardiac NDRL papers in the field and included all of them in this work. Cardiac NDRL papers are often published as nuclear medicine NDRL articles or with nuclear medicine and PET/CT NDRL papers. Few articles in the field focus only on reported cardiac NDRL. Therefore, I think we made a great effort and cited all the available information related to cardiac MPI, and we compared our data with the available, published, peer-reviewed NDRL data.

In regard to the second part of your observation, I have updated the introduction section to include some information about the variation between SPECT/CT protocols, taking into account the variation in radiation doses resulting from theses protocols.

Point 2: Does the intorduction provide sufficnet background and include all relevant refrences?

Respone 2: Thanks for providing your valuable comments. I have updated the introduction (background) to improve the quality of the work as follow" In clinical practice, a wide range of SPECT/CT and PET/CT protocols and radiopharmaceuticals can be used to perform the MPI scan while ensuring that a lower radiation dose is used on the patient [1,2]. The cardiac SPECT/CT and PET/CT MPI scan commonly consist of the stress protocol, which involves treadmill exercise or vasodilator drugs, and the rest protocol. The most common SPECT/CT MPI protocol conducted in one day, either starting with stress or rest, or two days protocols uses 99mTc-Tetrofosmin, 99mTc-Sestamibi, or thallium-201 (201Tl). In some circumstances, the patient undergoes only a stress SPECT/CT MPI and the stress result becomes normal; therefore, there is no need to do the rest protocol. Notably, doing the stress and rest MPI SPET/CT scan resulted in more radiation burden than doing only the stress SPECT MPI protocol [1,2,5]. Moreover, 201Tl is a good tracer for MPI; however, it has numerous limitations, such as a long half-life, high radiation exposure to the patient, relatively low injected activity, low signal-to-noise ratio, and, in some cases, suboptimal images particularly in overweight patients [1,2,5].”

Also, I added another point to imporve the quality of the work as follow” Furthermore, the literature review highlighted that variation in NDRL value for the administered activities related to the stress and rest -one day protocol range from 407 MBq and 740 MBq to 555 MBq and 1184 MBq for 99mTc-Tetrofosmin /Sestamibi respectively [8,11,12]. Regarding CT components, a few articles reflect the change of variation in NDRL related to the CT dose, for instance, the CTDIvol value range from 2.2 mGy and to the 36 mGy and the DLP value range from 36 mGy.cm to 380 mGy.cm [13-17].”

Point 3: Is the resarch design appropriate?

Response 3: I have updated the introduction and methodology sections to make sure the design of methodology section reflects the gap that I need to address in this paper. I have added some sentences to reflect the changes in radiation doses in the published international data at the end of the introduction section, as follows: “The literature review highlighted that variation in NDRL value for the administered activities related to the stressand rest one-day protocol which range from 407 MBq and 740 MBq to 555 MBq and 1184 MBq for 99mTc-teterofosmine/Sestamibi, respectively. Regarding CT components, a few articles reflect the change of variation in NDRL related to the CT dose; for instance, the CTDIvol value ranged from 2.2 mGy to 36 mGy and the DLP value ranged from 36 mGy.cm to 380 mGy.cm.”

I have updated the methodology section as follows: “The study recruited all adult Saudi patients who suffered from CVD and were eligible to undergo SPECT/CT or  PET/CT MPI scans between 2020 and 2021. Ethical approval (IRB log number: 21-204E) was acquired before commencing the study.The data were gathered from scans conducted on one SPECT/CT scanner and one PET/CT scanner in Riyadh, Saudi Arabia. Five SPECT/CT MPI scans were selected for this study: one-day stress/rest (99mTc-Sestamibi and 99mTc-Tetrofmise), two-day stress/rest (99mTc-Sestamibi and 99mTc-Tetrofmise), and stress/rest and reinjection 201TL. For PET/CT MPI scans, one stress/rest 82RB was selected for this study.”

Point 4: Are the result clerarly presntented?

I made some minor modifications to clarify the value of the SPECT/CT MPI protocols.